# How small-molecule inhibitors of dengue-virus infection interfere with viral membrane fusion

**Luke H Chao[1†‡], Jaebong Jang[1,2], Adam Johnson[1], Anthony Nguyen[1], Nathanael S Gray[1,2], Priscilla L Yang[3], Stephen C Harrison[1,4]\***

[1]Department of Biological Chemistry and Molecular Pharmacology, Harvard Medical School, Boston, United States; [2]Department of Cancer Biology, Dana-Farber Cancer Institute, Boston, United States; [3]Department of Microbiology and Immunobiology, Harvard Medical School, Boston, United States; [4]Howard Hughes Medical Institute, Harvard Medical School, Boston, United States

**\*For correspondence:**
harrison@crystal.harvard.edu

**Present address:** [†]Department of Molecular Biology, Massachusetts General Hospital, Boston, United States; [‡]Department of Genetics, Harvard Medical School, Boston, United States

**Competing interests:** The authors declare that no competing interests exist.

**Abstract** Dengue virus (DV) is a compact, icosahedrally symmetric, enveloped particle, covered by 90 dimers of envelope protein (E), which mediates viral attachment and membrane fusion. Fusion requires a dimer-to-trimer transition and membrane engagement of hydrophobic 'fusion loops'. We previously characterized the steps in membrane fusion for the related West Nile virus (WNV), using recombinant, WNV virus-like particles (VLPs) for single-particle experiments (Chao et al., 2014). Trimerization and membrane engagement are rate-limiting; fusion requires at least two adjacent trimers; availability of competent monomers within the contact zone between virus and target membrane creates a trimerization bottleneck. We now report an extension of that work to dengue VLPs, from all four serotypes, finding an essentially similar mechanism. Small-molecule inhibitors of dengue virus infection that target E block its fusion-inducing conformational change. We show that ~12–14 bound molecules per particle (~20–25% occupancy) completely prevent fusion, consistent with the proposed mechanism.
DOI: https://doi.org/10.7554/eLife.36461.001

## Introduction

Flaviviruses, a family that includes dengue, tick-borne encephalitis, West Nile, and Zika viruses, are a group of mosquito-borne pathogens of substantial interest for vaccine and small molecule therapeutic development (*Diamond and Pierson, 2015*; *Heinz and Stiasny, 2012*). The envelope glycoprotein (E) is the membrane fusogen required for cell entry and infection (*Harrison, 2015*). E comprises three beta-sheet rich domains: domain I (DI) is a central beta barrel that organizes the rest of the subunit; domain II (DII: an extension emanating from DI) bears a hydrophobic fusion loop at its distal tip; domain III (DIII), with an immunoglobulin-like fold, connects to the C-terminal 'stem' and transmembrane helical hairpin (*Rey et al., 1995*). The conserved, fusion-inducing conformational rearrangements these proteins undergo when exposed to reduced pH, from a pre-fusion dimer to a post-fusion trimer, offer an opportunity to exploit the common structural features of all flavivirus E proteins to find viral entry inhibitors (*Figure 1*).

In a crystal structure of the dengue serotype 2 E dimer, a molecule of β-octyl glucoside (β-OG), used in the preparation, was present in a pocket between DI and DII (*Modis et al., 2003*). This pocket closes up during the fusogenic conformational change (*Modis et al., 2004*), and the observation that a small molecule can bind there suggested that it might be a good target for an entry inhibitor. Indeed, a screen originally designed to detect small molecules that interfere with another step in the conformational transition yielded a series of cyanohydrazone compounds that bind a

soluble DI-DII fragment (*Schmidt et al., 2012*). These compounds block fusion in vitro and infection in cell culture; they bind the E protein on the virion surface before the virus attaches to a cell. The most potent of them inhibit viral infectivity with IC$_{90}$ values in the single-digit micromolar range (*Schmidt et al., 2012*). Recent studies of the effects of mutation on members of several chemically distinct series of inhibitors, including the cyanohydrazones, confirm that the β-OG pocket is indeed their binding site (*de Wispelaere et al., 2018*).

In previous work on the mechanism of membrane fusion by West Nile virus (WNV) E, we used single-particle fusion measurements with WNV virus-like particle (VLPs) to outline steps in the transition leading to membrane merger (*Chao et al., 2014*). We could interpret the observed, pH-dependent kinetics with a model in which trimerization of E into an extended intermediate, with the three fusion loops inserted into the target membrane, becomes a kinetic bottleneck in progression to hemifusion, because hemifusion requires collapse to the postfusion conformation of at least two *adjacent* trimers in a ~30 monomer contact zone between a virus particle and the membrane with which it is fusing (*Figure 1*). Stochastic simulations gave estimates for the rates of various steps. In the present work, we have extended the analysis to all four dengue virus (DV) serotypes, using VLPs as in our studies of WNV fusion. We also use the single-particle approach to examine inhibition by compounds in the cyanohydrazone series and to estimate the number of small-molecule inhibitors per particle needed to block fusion.

## Results

### Single-particle measurement of dengue VLP hemifusion kinetics

VLPs for the four DV serotypes were prepared by expression in 293 T cells, essentially as described for WNV VLPs (*Chao et al., 2014*). Expression was at 28°C rather than 37°C, as the latter yielded particles with substantially lower fusion activity. We measured the pH dependence of bulk hemifusion and found sharply sigmoidal curves with inflection pH of 6.1 for the DV4 VLPs and ~5.4 for those of DV1, DV2 and DV3 (*Figure 2A*). We used total internal reflection fluorescence microscopy as

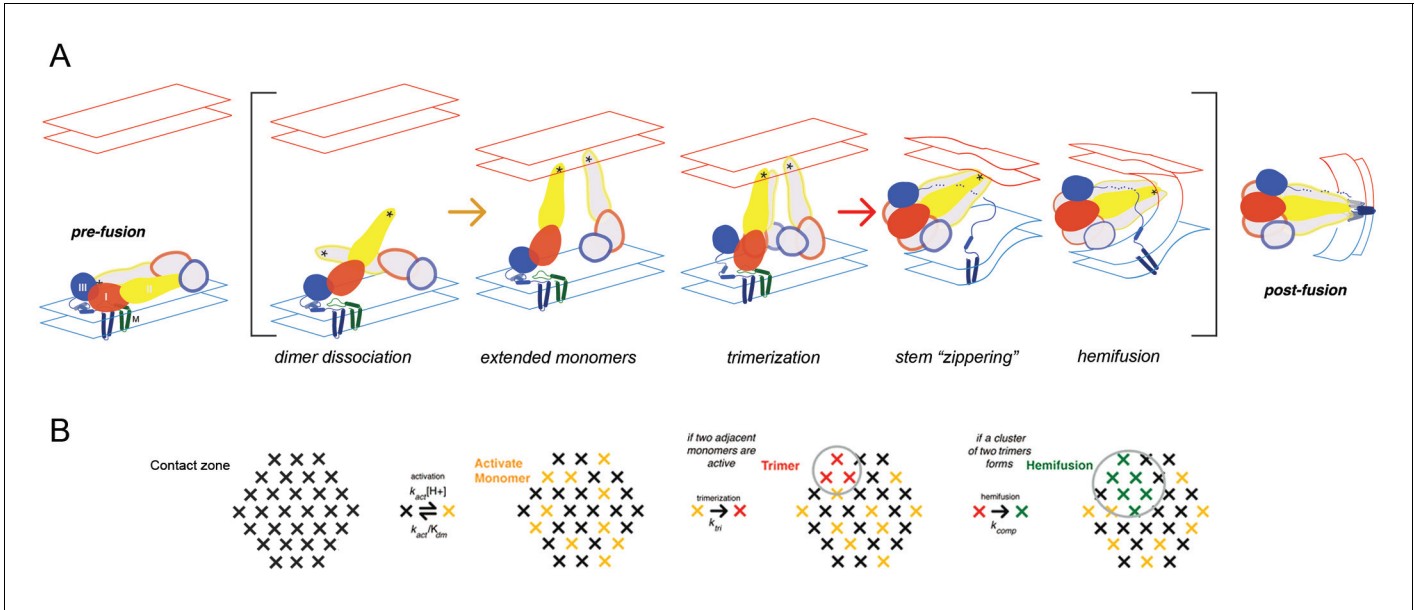

**Figure 1.** Flavivirus membrane fusion (A) Steps in the fusion transition. Starting point and end point represent known structures; steps in brackets inferred from experimental evidence as summarized in (*Chao et al., 2014*). (B) Scheme for simulation of fusion reaction. Array of crosses represents a contact zone with 30 monomers.

DOI: https://doi.org/10.7554/eLife.36461.002

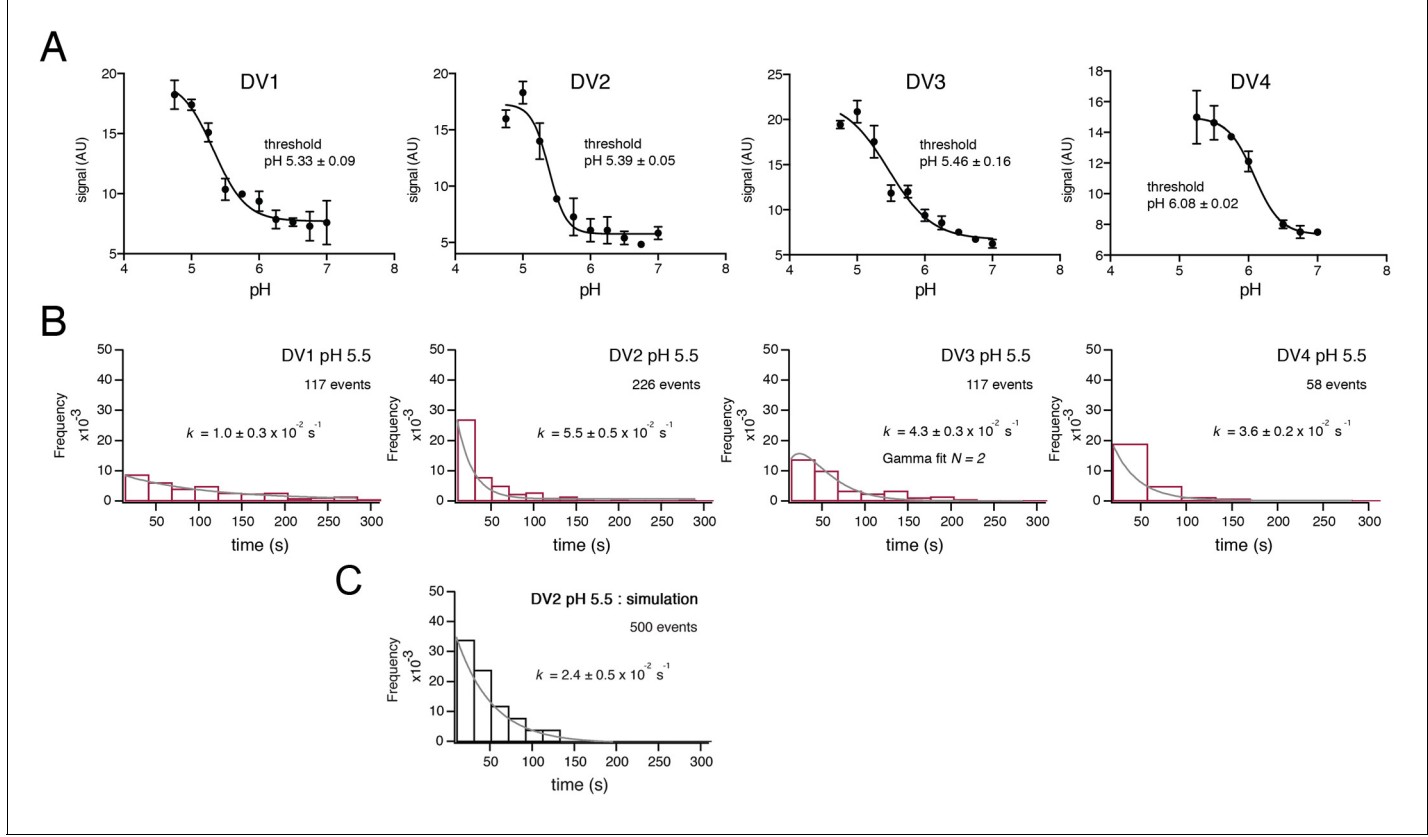

**Figure 2.** Fusion measurements. (**A**) Fusion (with liposomes, in bulk solution) for VLPs of the four DV serotypes. The fluorescence from membrane-incorporated DiD is shown as a function of pH. Hemifusion (or fusion) at low pH causes dequenching of the VLP-incorporated fluorophore. (**B**) Histograms of single-particle fusion dwell times (between lowering of pH and observed dequenching) at pH 5.5 for each of the four DV serotypes. Curves show fit with a single exponential (DV1, DV2, and DV4) or with a gamma distribution, N = 2 (DV3). (**C**) Results of a simulation with parameters as described in the text. Compare with experimental data for DV2 fusion in the panel immediately above it.
DOI: https://doi.org/10.7554/eLife.36461.003

The following figure supplement is available for figure 2:

**Figure supplement 1.** Simulation diagram.
DOI: https://doi.org/10.7554/eLife.36461.004

described previously (*Chao et al., 2014*; *Floyd et al., 2008*; *Ivanovic et al., 2013*; *Kim et al., 2017*) to determine single-particle dwell time distributions for hemifusion at pH 5.5 (*Figure 2B*), with dequenching of DiD-labeled VLPs to mark the moment of hemifusion with the supported lipid bilayer. We incorporated into the target bilayer a pseudo-receptor, to uncouple membrane attachment from exposure of the fusion loop triggered by the pH drop. The receptor was either the lectin domain of DC-SIGN-R linked through a histidine tag to a NiNTA-headgroup lipid or a similarly linked, antigen-binding fragment (Fab) from antibody 1AID-2, specific for DV2 domain II (*Chao et al., 2014*; *Lok et al., 2008*; *Tassaneetrithep et al., 2003*). We found no receptor-dependence of hemifusion dwell times; the data shown are with the DC-SIGN-R.

The dwell-time distributions in *Figure 2B* show some differences among the four serotypes. The exponential used to fit the distributions for DV1, DV2 and DV4 VLPs gives an effective rate constant for a first-order 'rate-limiting step'; the best formal fit for DV3 includes two successive or parallel steps. The bulk fusion vs. pH curve is also less sharply sigmoidal than are the others. Although the inflection pH measured in bulk is similar for DV1, DV2 and DV3, the effective first-order rate of fusion for DV1 is slower. We cannot, however, with our current data relate these differences to functional differences among the isolates.

## Small-molecule inhibition

The compound 3-110-22 (*Figure 3*) inhibits DV2 infection with $IC_{90}$ of 0.7 µM (*Schmidt et al., 2012*). The $IC_{50}$ for DV2 VLP fusion, measured in bulk, was 1–2 µM (*Figure 3A*). We used a concentration of 1 µM to study the effect of 3-110-22 on the dwell-time distribution for single-particle fusion at pH 5.5 (*Figure 3B*). Comparison of *Figure 3B* with the DV2 VLP panel in *Figure 2B* shows that presence of the inhibitor has spread the distribution and introduced a clear rise and fall, suggesting that the inhibitor has retarded a step other than the one that is effectively rate-limiting in its absence.

To relate single-particle hemifusion dwell-time distribution and small-molecule occupancy, we conjugated a fluorescent probe (Alexa-555) to 3-110-22 at the 4-position of the benzene ring (see Materials and methods). We confirmed that the modified form of the inhibitor was active by recording bulk hemifusion inhibition and found its $IC_{50}$ to be about 10- to 20-fold higher that of the unmodified compound; the strong tendency of these hydrophobic compounds to associate in solution reduces the effective monomer concentration, and the actual difference may be smaller (*Figure 3C*). We also showed that 3-110-22 competed for binding with the modified inhibitor, indicating that they both interact at the same site (*Figure 3D*).

We incubated DV2 VLPs with 1 µM Alexa-555/3-110-22, removed excess compound with a desalting column (*Schmidt et al., 2012*), and measured single-particle hemifusion events. We determined inhibitor occupancy for each particle using a calibrated intensity in the Alexa-555 channel (see Materials and methods) and correlated the occupancy with hemifusion fate of that particle. At pH 5.5 under the conditions of our experiment, the overall yield of hemifusion in the absence of inhibitor was ~25% (i.e. the percent of total particles in the observation field that ultimately fused). Particles with more than about 12 bound inhibitors failed to fuse at all (*Figure 4*). We did not detect

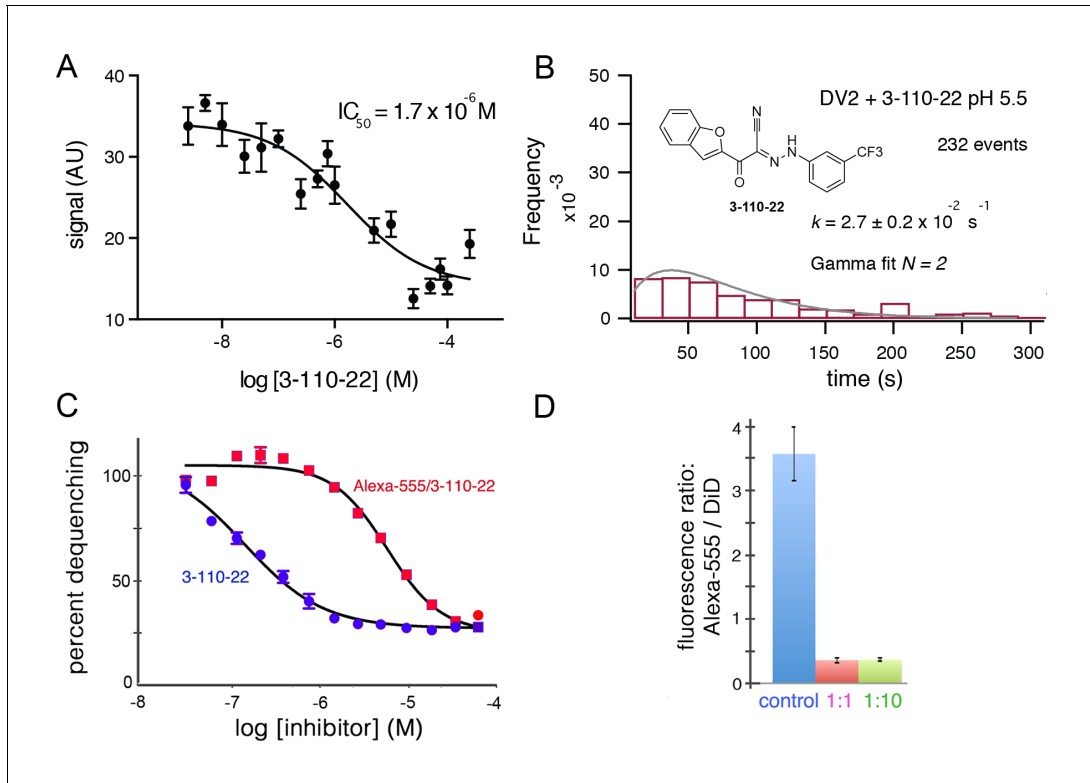

**Figure 3.** Inhibition of DV2 VLP fusion by 3-110-22. (**A**) Fluorescence dequenching as a function of inhibitor concentration. (**B**) Single-particle dwell-time distribution at pH 5.5 in the presence of 1 µM inhibitor. (**C** and **D**) Inhibition of DV2 VLP fusion by Alexa-555/3-110-22. (**C**) Fluorescence dequenching as a function of inhibitor concentration. Percent dequenching calculated with 100% as DiD dequenching with no added inhibitor and 0% as dequenching with no pH drop. Error bars are SEM, n = 3. (**D**). Single-particle binding intensity for Alexa-555/3-110-22 in the presence of varying molar ratios of underivatized 3-110-22 (none, 1:1, 1:10). Error bars: SEM; n = 373, 370, 382 for the three sets of measurements, respectively.
DOI: https://doi.org/10.7554/eLife.36461.005

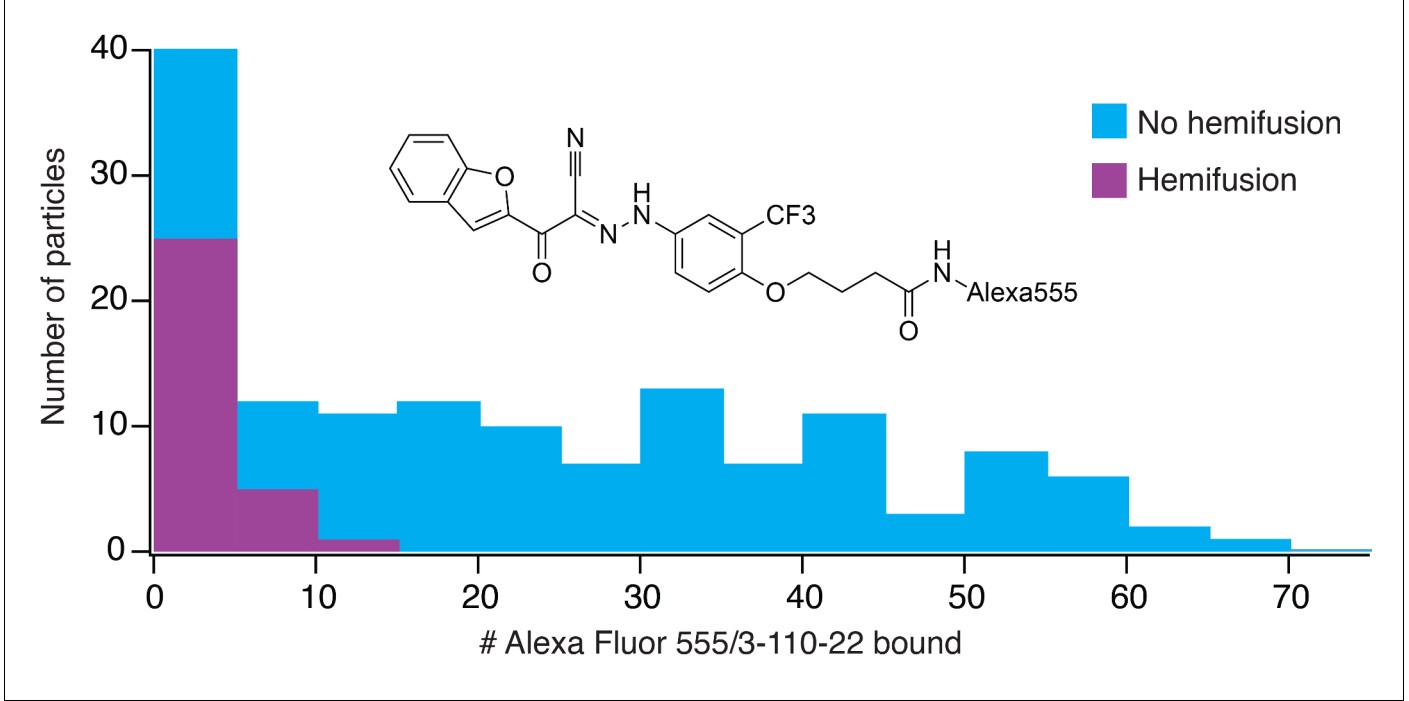

**Figure 4.** Frequency of hemifusion (measured as DiD fluorescence dequenching) as a function of number of bound Alexa-fluor-555/3-110-22 molecules. Histogram shows the number of particles with a particular number of bound fluorescent inhibitor molecules, in bins of five. The number of particles in a bin that proceeded to hemifusion in the time of the experiment is shown as the height of the purple bar; the number of non-fusing particles is the total height of the bar (blue).

DOI: https://doi.org/10.7554/eLife.36461.006

particles with more than about 60 copies of Alexa-555/33-110-22, showing that binding saturated at an occupancy consistent with a large fraction of 'small VLPs' (~60 E subunits) in the preparation. Assuming that the contact zone between the particle and the membrane includes about 30 E subunits (i.e. about half a small VLP) and that binding at one site is independent of the occupancy of any other site, then ~6 bound inhibitors in the contact should block fusion completely. The percent of uncleaved prM, estimated from Western blots, was between 10 and 20% for various preparations. Residual uncleaved prM is usually found in a single cluster on virus particles (*Plevka et al., 2011*); it should therefore be absent from the contact zone with the target membrane, to which the prM associated E cannot attach. If there were no binding of the fluorescent inhibitor to prM-associated E, our estimate for the number of bound inhibitors per fully mature particle needed to prevent fusion completely would be about 14, rather than 12, or 7 in the contact zone, rather than 6. Thus, presence of a small fraction of uncleaved prM on the particles in our experiments would have had at most a relatively small effect on our determination of the inhibitory threshold.

## Simulation of DV2 fusion

We modeled the conformational steps preceding hemifusion with a stochastic simulation, structured like the scheme we used for simulations of West Nile virus hemifusion (*Figure 2C*) (*Chao et al., 2014*). A hexagonal array of 30 E protomers represents the contact patch between viral and cell membranes. Each E subunit transits between an inactive state (in a dimer) and an activated monomer, which can trimerize when two other adjacent monomers are activated. Hemifusion occurs upon formation of two adjacent trimers. Dimer pairs are explicitly defined in the simulation, and a cooperativity factor increases the probability of activation when the dimer partner is already activated. The threshold for trimerization was set to pH 5.4, the bulk threshold for hemifusion, based on our previous finding that these two thresholds are the same for WNV (*Chao et al., 2014*). The assumption of reversibility for the DV2 E dimer-monomer transition comes from dynamic light scattering measurements performed on Kunjin virus (*Chao et al., 2014*). The DV2 simulation produced a dwell time

histogram at pH 5.5 that could be fit with a single exponential decay (*Figure 2C*). In our simulation, the yield for particles achieving hemifusion (defined as the total number of fusion events in a time window divided by the total number of identified particles) was ~25%, as observed experimentally.

## Discussion

The single-particle fusion kinetics for VLPs of all four dengue serotypes are qualitatively similar to those of WNV (*Chao et al., 2014*). Moreover, explicit simulation of DV2 fusion corresponds to direct experimental observation, for values of the simulation parameters close to those used to fit corresponding data for WNV. The E proteins of WNV and the four serotypes of dengue virus are closely related to each other, and their comparable in vitro fusion properties support the robustness of both our experimental design and our fusion model.

Inhibition of dengue virus fusion by 33-110-22, shown previously in bulk assays, confirms its likely mode of action in blocking infection (*Schmidt et al., 2012*). The inhibitor and its analogs bind E on the virion surface before the particle attaches to a cell, at a site on each of the E monomers that closes up during the transition from prefusion dimer to postfusion trimer (*Modis et al., 2003*, *2004*; *Schmidt et al., 2012*; *de Wispelaere et al., 2018*). The bound molecules appear to fix the E protein in the conformation present in the prefusion dimer, blocking progression along the fusogenic pathway. We have shown that the assay originally used to identify the cyanohydrzone inhibitors probably worked because these molecules lock the hinge in a dimer-like configuration, even in the context of a fusion-intermediate trimer (*Klein et al., 2013*). Synthesis of a fluorescent derivative has now enabled us to examine its effects on a single-particle basis and hence to derive not only a bulk inhibitory $IC_{50}$ but to determine the fusion efficiency as a function of the number of molecules bound.

Most of the particles in our preparations were small, ~60 subunit VLPs. As we concluded above from the data in *Figure 4*, the parameter likely to be relevant to small-molecule inhibition is the number of E proteins in the contact zone between particle and target membrane, estimated here, as in our previous work, as ~30. The experimental results show that about 6–7 inhibited monomers in a 30-monomer contact zone (a fractional occupancy of about 0.2) will be enough to prevent fusion completely (see *Figure 4*). Our previous work with WNV VLPs showed that virions, virion-sized (180-subunit) VLPs, and small VLPs have indistinguishable fusion kinetics, including their pH dependence (*Chao et al., 2014*). Therefore, about 35 inhibitors per dengue virion (i.e. about 20% occupancy) should block any detectable fusion, assuming that for complete inhibition the threshold density of bound inhibitor is the same for the two size classes. An important inference is that complete (or nearly complete) inhibition should occur at concentrations substantially lower than the Kd for an individual binding event.

In the simulation scheme in *Figure 1B*, modeling the effect of inhibitor binding to a monomer, at any step in the fusogenic conformational change, by inactivating at random seven monomers does not lead to full inhibition without further restrictions on the model. That is, there are ways of distributing inactivated monomers within the 30-monomer contact zone that still allow two neighboring, activated trimers to form, although in general more slowly than if all monomers could participate (see *Figure 2—figure supplement 1*). We suggest two potential explanations: (i) failure of an inhibitor-free trimer to associate with the target membrane, leading to an abortive fold-back with insertion of the fusion loops into the virion membrane instead; (ii) dimer cooperativity, such that binding of inhibitor to one dimer partner inactivates (with some probability) the other, for example, by inhibiting dimer dissociation, thereby preventing the unliganded partner from forming an active trimer with other neighbors. The former mechanism would add a dead-end side reaction to the trimer zippering step in *Figure 1*. It is precisely analogous to the mechanism that accounts for a closely related observation in studies of influenza virus fusion (*Ivanovic and Harrison, 2015*; *Otterstrom et al., 2014*). The latter mechanism would generate more effectively inhibited monomers than actual bound inhibitor molecules. Both inspection of diagrams such as the one in *Figure 2—figure supplement 1* and modified simulations with randomly inactivated monomers in the contact zone show that a total of eight or nine inactive monomers in the contact zone – just one or two in addition to the seven with bound inhibitor – would be enough to rule out formation of two adjacent, active trimers. Thus, the abortive trimer model would require only a low frequency of dead-end events, with minimal effect on fusion yield or kinetics in the inhibitor-free case (see also the similar

conclusion for influenza fusion: *Ivanovic and Harrison, 2015*), and the dimer cooperativity model would require a relatively low probability for inactivation of the unliganded dimer partner (e.g. only partial inhibition of dimer dissocation). The geometrical idealizations inherent in the current simulation model do not allow us to distinguish between these alternatives.

## Materials and methods

### Virus-like particles (VLPs)

Dengue virus-like particles were produced from a stable, mycoplasma-free 293 T cell line (obtained originally from the Broad Institute (*Luo et al., 2008*), maintained in our laboratories since then, and not further characterized) transfected with the pVRC8400 expression vector bearing a structural cassette containing a codon-optimized version of the prM-E sequence from the DV 1 clone 45AZ5, DV two clone Harvard/BID-V2992/2009, DV three isolate IN/BID-V2417/1984, or DV4 isolate TVP/360. A tissue plasminogen activator signal sequence preceded prM-E. The mycoplamsa-free, adherent cells were grown at 28°C in Gibco FreeStyle 293 medium (Life Technologies, Grand Island, NY). The DV VLPs in the medium were harvested after 2 days, clarified from debris by low-speed centrifugation, and precipitated with polyethylene glycol 8000. Following resuspension in buffer containing 20 mM tricine (N-(2-Hydroxy-1,1-bis(hydroxymethyl)ethyl)glycine) pH 7.8, 140 mM NaCl and 0.005% Pluronic F-127, VLPs were purified over an Optiprep density gradient (SW41 rotor, 34,000 rpm, 4°C, 2 hr. 20 min.) with 55−45−35−30−25−20−10% steps. We collected the band between the 35 and 30% densities and found this material to contain particles 35 and 50 nm in diameter as assessed by cryo- and negative-stain electron microscopy (*Allison et al., 2003*). The percent of uncleaved prM, estimated from Western blots, was between 10 and 20% for various preparations. Particle membranes were labeled with DiD (1,1'-dioctadecyl-3,3,3',3'-tetramethylindodicarbocyanine perchlorate) at ~20 µM or 20-fold the protein concentration. Excess dye was removed using NAP-10 desalting column (GE Healthcare, United Kingdom).

### Small molecule synthesis

Synthesis of Alexa fluor 555 conjugate began with phenol installation on 3-110-22 at the 4-position of the benzene ring according to the procedure previously reported (*Schmidt et al., 2012*). A primary amine with a four-carbon linker was attached to the phenol by etherification and deprotection. Amidation between the primary amine and the commercially available Alexa fluor 555 NHS ester then generated an Alexa fluor 555 conjugated 3-110-22, which was confirmed by $^1$H NMR.

### General SI-Chemistry

All reactions were monitored by LC/MS (Waters 2998 Photodiode Array Detector, Waters SQ detector 2, Waters 515 HPLC pump, Waters 2545 Binary Gradient Module, Waters System Fluidics Organizer and Waters 2767 Sample Manager) using a SunFireTM C18 column (4.6 × 50 mm, 5 µm particle size): solvent gradient = 80% A at 0 min, 1% A at 5 min; solvent A = 0.035% TFA in Water; solvent B = 0.035% TFA in MeOH; flow rate: 1.5 mL/min. Reaction products were purified by flash column chromatography using CombiFlashRf with Teledyne Isco RediSepRf High Performance Gold or Silicycle SiliaSepTM High Performance columns (4, 12, 24, 40, or 80 g) and a Waters HPLC system using SunFireTM Prep C18 column (19 × 100 mm, 5 µm particle size): solvent gradient = 80% A at 0 min, 5% A at 25 min; solvent A = 0.035% TFA in water; solvent B = 0.035% TFA in MeOH; flow rate: 25 mL/min. The purity of all compounds was greater than 95% as analyzed by LC/MS (see above). Chemical shifts are reported in parts per million (ppm, δ) downfield from tetramethylsilane (TMS). Coupling constants (*J*) are reported in Hz. Spin multiplicities are described as br (broad), s (singlet), d (doublet), t (triplet), q (quartet) and m (multiplet).

**Chemical structure 1.** Addition of four-carbon linker with primary amine to 3-110-22

DOI: https://doi.org/10.7554/eLife.36461.007

Compound 1 was prepared from 4-amino-2-(trifluoromethyl)phenol as described (*Schmidt et al., 2012*). To a solution of compound 1 (250 mg, 0.67 mmol) in *N,N*-dimethylformamide (2 mL) were added *tert*-butyl (4-bromobutyl)carbamate (101 mg, 0.40 mmol) and potassium carbonate (139 mg, 1.01 mmol). After stirring at 70°C for 6 hr, the reaction mixture was cooled to room temperature, diluted with EtOAc and washed five times with water. The organic layer was dried over anhydrous sodium sulfate, filtered, and concentrated under reduced pressure. The residue was purified by flash column chromatography (DCM: MeOH = 90: 10) to afford compound 2 as a yellow solid (200 mg, 91%). m/z: 567.13 [M + Na]$^+$; $^1$H NMR (600 MHz, CD$_3$OD) $\delta$ 7.92 (s, 1 hr), 7.85–7.82 (m, 1 hr), 7.80 (d, *J* = 7.6 Hz, 1 hr), 7.77–7.73 (m, 1 hr), 7.63 (d, *J* = 8.8 Hz, 1 hr), 7.54 (t, *J* = 7.6 Hz, 1 hr), 7.37 (t, *J* = 7.3 Hz, 1 hr), 7.29 (d, *J* = 9.4 Hz, 1 hr), 4.14 (t, *J* = 6.2 Hz, 2 hr), 3.11 (t, *J* = 6.8 Hz, 2 hr), 1.87–1.81 (m, 2 hr), 1.71–1.64 (m, 2 hr) 1.43 (s, 9 hr).

**Chemical structure 2.** Conjugation of Alexa fluor 555 with activated 3-110-22.
DOI: https://doi.org/10.7554/eLife.36461.008

Step 1: To a solution of compound 2 (200 mg, 0.37 mmol) in CH$_2$Cl$_2$ (4 mL) was added trifluoro-acetic acid (1 mL). After stirring for 1 hr, the reaction mixture was diluted with CH$_2$Cl$_2$ and alkalinized with saturated sodium bicarbonate. The organic layer was separated, and the aqueous layer was extracted five times with CH$_2$Cl$_2$. The combined organic layer was dried over anhydrous sodium sul-fate, filtered and concentrated under reduced pressure to give *N*-Boc deprotected precursor as a yellow oil (152 mg, 92%) m/z: 445.07 [M + 1]+.

Step 2: To a solution of compound 2 (2.7 mg, 0.006 mmol) in *N,N*-dimethylformamide (0.5 mL) were added DMAP (0.6 mg, 0.0048 mmol) and Alexa fluor 555 NHS ester (ThermoFisher Scientific, m.w. ~1250, 5 mg, 0.004 mmol). The resulting mixture was stirred at room temperature for 3 hr, diluted with DMSO and the product purified by prepHPLC to give compound three as a red solid (m.w. ~1579, 1.5 mg, 24%).

## Single particle assay

Single particle data were collected as previously described (*Chao et al., 2014*). Briefly, glass cover-slips were cleaned by sonication in '7X' detergent, 1M potassium hydroxide, acetone and ethanol, and dried for 1 hr at 100°C. Polydimethylsiloxane (PDMS) flow cells with 0.5 mm wide and 70 µm high channels (five per cell) were bonded to plasma-treated glass. Teflon FEP tubing (0.2 mm, Upchurch Scientific) connected an Eppendorf tube with solution to the channel, and Intramedic poly-ethylene tubing (0.76 mm) connected the channel to a syringe pump (Harvard Pump 11; Harvard Apparatus, Holliston, MA).

Liposomes for preparing planar bilayers contained 1-palmitoyl-2-oleoyl-sn-glycero-3-phosphoe-thanolamine (POPE), 1-oleoyl-2-palmitoyl-sn-glycero-3-phosphocholine (POPC), cholesterol, and 1,2-dioleoyl-sn-glycero-3-phosphocholine (DOPC), 1,2-dioleoyl-sn-glycero-3-phosphoethanolamine-N-(carboxyfluorescein) (FL-PE) and 1,2-dioleoyl-sn-glycero-3-[(N-(5-amino-1-carboxypentyl)iminodiace-tic acid)succinyl] (Ni-NTA DOGS) (Avanti Polar Lipids, Alabaster, AL) in a ratio of 4:2:2:2:0.02:1%. Lip-osomes at 10 mg/ml were extruded through a 200 nm pore-size polycarbonate membrane filter. Liposomes were loaded into the flow cell, and the flow stopped to allow bilayers to form. We per-formed fluorescence recovery after photobleaching experiments to confirm the fluidity of the bilayer. Unattached liposomes were washed away, and 1A1D-2 Fab or the lectin domain of DCSIGN-R, with a C-terminal His6 tag, was introduced at 50 nM for 2 min. 1A1D-2 Fab was produced from a stable 293T line expressing both heavy and light chains from the pVRC8400 vector, purified by Ni-affinity chromatography and S200 size-exclusion chromatography. DCSIGN-R was expressed from Hi-5 cells infected with recombinant baculovirus. Labeled virus particles were loaded onto the pseudo-recep-tor decorated bilayer. To initiate fusion, we introduced acetate buffer (100 mM sodium acetate, pH 5.0–5.5) or MES (100 mM, pH 5.75–6.25), with 140 mM sodium chloride and 0.005% Pluronic F-127.

## Fluorescence measurements

End-point bulk fusion data were collected using a GE Amersham Typhoon plate reader at 633 and 670 nm excitation and emission wavelengths respectively in 96-well clear-bottom plates with 2 mg/ml final lipid concentration (200 nm liposomes prepared as described above). VLPs were prepared and labeled with DiD as previously described (*Chao et al., 2014*).

Bulk liposome fusion data were collected on a PTI (Photon technology International, Edison, NJ) 814 Fluorimeter, with 648 and 669 nm excitation and emission wavelengths respectively, using a Cole–Parmer digital polyStat temperature controlled thermo-jacket at 2 Hz over 10 min and at 0.2 mM final lipid concentration (200 nm liposomes prepared as described above).

Single-particle fusion data were collected on an inverted Olympus IX71 fluorescence microscopy with a high numerical aperture objective (60×, N.A. = 1.3). VLPs were illuminated with 488 and 640 nm Coherent (Wilsonville, OR) lasers. A custom-fabricated water-chilled temperature collar (Bioptecs, Butler, PA) was fitted on the objective turret. Each time-lapsed fluorescence video was recorded at 1 Hz for 300 s using 3i Slidebook software. Data were analyzed using Igor and MatLab. $IC_{50}$ was determined by non-linear regression.

The single-fluorophore fluorescence intensity was calibrated by drying buffer containing the labeled small molecule onto a glass slide and monitoring single step photobleaching events under identical imaging conditions as in the experiment.

## Competition assay

We used TIRF microscopy, as described (*Chao et al., 2014*). Coverslips were sonicated in ethanol and water and then glow discharged prior to use. VLPs were dye labeled as in fusion experiments, but at 1:10 the concentration of DiD as we did not require quenching; the concentration of VLPs was five times higher than used for the bulk fusion assay. Labeled VLPs were mixed with 1 µM fluorescent inhibitor for five mins before loading onto the TIRF. For competition assays with unlabeled inhibitor, DiD labeled VLPs were first incubated with the 3-110-22 for 5 mins at 1 or 10 µM, followed by addition of Alexa555/3-110-22 for five mins.

Calcuations were made by densitometry of the VLP spots as identified in ImageJ. The spots were found with the VLP fluorescent channel (DiD) and then both channels were measured, background subtracted, and the ratio of Alexa555 to DiD determined.

## Simulations

Matlab code, modified from our previous work (*Chao et al., 2014*), was used with parameters optimized against experimentally measured values for DV2. Code is publicly available at https://github.com/Harrison-Lab/Flavivirus (*Harrison-Lab, 2018*; copy archived at https://github.com/elifesciences-publications/Flavivirus).

# Acknowledgements

We thank Ilya Kuesters and T Kirchhausen (Boston Children's Hospital) for assistance with TIRF microscopy and access to facilities. LHC was a Frederic M Richards Fellow of the Jane Coffin Childs Memorial Fund for Medical Research and Charles King Trust Fellow. AN was supported by the Howard Hughes Medical Institute EXROP program. SCH is an investigator in the Howard Hughes Medical Institute. We acknowledge support from NIH grants CA13202 and AI109740 (Center for Excellence in Translational Research).

# Additional information

### Funding

| Funder | Grant reference number | Author |
|---|---|---|
| National Cancer Institute | CA13202 | Stephen C Harrison |
| National Institute of Allergy and Infectious Diseases | AI109740 | Stephen C Harrison |

| Howard Hughes Medical Institute | Stephen C Harrison |
| Charles A. King Trust | Luke H Chao |
| Jane Coffin Childs Memorial Fund for Medical Research | Luke H Chao |

The funders had no role in study design, data collection and interpretation, or the decision to submit the work for publication.

## Author contributions
Luke H Chao, Conceptualization, Data curation, Software, Formal analysis, Supervision, Investigation, Methodology, Writing—original draft, Writing—review and editing; Jaebong Jang, Data curation, Methodology, Writing—review and editing; Adam Johnson, Data curation, Formal analysis, Investigation, Methodology, Writing—review and editing; Anthony Nguyen, Data curation; Nathanael S Gray, Priscilla L Yang, Conceptualization, Supervision, Writing—review and editing; Stephen C Harrison, Conceptualization, Formal analysis, Writing—original draft, Writing—review and editing

## Author ORCIDs
Luke H Chao (iD) https://orcid.org/0000-0002-4849-4148
Nathanael S Gray (iD) https://orcid.org/0000-0001-5354-7403
Priscilla L Yang (iD) http://orcid.org/0000-0001-7456-2557
Stephen C Harrison (iD) http://orcid.org/0000-0001-7215-9393

## Decision letter and Author response
Decision letter https://doi.org/10.7554/eLife.36461.013
Author response https://doi.org/10.7554/eLife.36461.014

# Additional files
## Supplementary files
• Transparent reporting form
DOI: https://doi.org/10.7554/eLife.36461.009

## Data availability
Simulation software deposited at GitHub (flavi_hemi_20180221.m: Harrison SC, 2018, https://github.com/Harrison-Lab/Flavivirus/blob/master/flavi_hemi_20180221.m, 8eee0aae). Copy archived at https://github.com/elifesciences-publications/Flavivirus.

The following datasets were generated:

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
