## [Decision Letter]

Thank you for submitting your article "How small-molecule inhibitors of dengue-virus infection interfere with viral membrane fusion" for consideration by *eLife*. Your article has been reviewed by three peer reviewers, and the evaluation has been overseen by a Reviewing Editor and Randy Schekman as the Senior Editor. The reviewers have opted to remain anonymous.

The reviewers have discussed the reviews with one another and the Reviewing Editor has drafted this decision to help you prepare a revised submission.

In this Research Advance article, Chao et al. extend the work they published in a research article in *eLife* in 2014 on the mechanism and dynamics of membrane fusion of West Nile (WN) virus-like particles to all four dengue virus serotypes. As in the original study, membrane fusion is assayed with liposomes in bulk, in and a single particle assay with a planar lipid bilayer installed in customized microfluidic cells imaged by TIRF microscopy. The 2014 study is extended further with experiments examining the mechanism of inhibition by a small hydrophobic compound, 3-110-22, previously identified as inhibitors of the fusogenic conformational transition in dengue virus. The authors then determined the minimum number of inhibitor molecules per virus like particle (VLP) to completely prevent fusion. The observed minimum number per VLP (15) along with a simulation suggests a model where the inhibitor interferes with the transition from dimer to monomer of the envelope protein.

All referees agree that your study constitutes an interesting extension of your previous work to a medically more relevant virus, confirming that the stochastic simulations developed previously indeed produces meaningful results. Thus, the referees suggest in principle publication provided that the following two issues are addressed during revision:

1) Most importantly, a possible caveat is that the authors have used 293T cells to produce the VLPs. A previous study reported that dengue virions produced in 293T cells yield near 70% uncleaved prM (Dejnirattisai et al., Nature Immunology 2015), and the presence of prM inactivates its E partner for fusion. In the previous publication on WNV, the authors had explicitly verified that uncleaved prM was only a minimal proportion of the particles. It seems important that they quantify here the amount of uncleaved prM, and if its contribution is important, it would require to explicitly model its presence in their simulations.

2) Another concern is regarding the location of the 3-110-22 binding site. The authors show that 3-110-22 and the fluorescently labeled analog compete for the same binding site, which is "presumed" to be the detergent binding pocket of the E protein. Based on the fusion simulation data and on the maximum number of fluorophore per VLP the compound is proposed to bind to the E protein dimer and inhibit the dimer to monomer transition as opposed to other steps downstream in the fusion reaction. Ultimately, a structure of the bound inhibitor would be of interest, but this is outside the scope of this Research Advance. In the absence of such a structure (or structure/function mutagenesis), a more in depth discussion of the likely binding site and likely mechanism of inhibition would be desirable.

---

## [Author Response]

All referees agree that your study constitutes an interesting extension of your previous work to a medically more relevant virus, confirming that the stochastic simulations developed previously indeed produces meaningful results. Thus, the referees suggest in principle publication provided that the following two issues are addressed during revision:1) Most importantly, a possible caveat is that the authors have used 293T cells to produce the VLPs. A previous study reported that dengue virions produced in 293T cells yield near 70% uncleaved prM (Dejnirattisai et al., Nature Immunology 2015), and the presence of prM inactivates its E partner for fusion. In the previous publication on WNV, the authors had explicitly verified that uncleaved prM was only a minimal proportion of the particles. It seems important that they quantify here the amount of uncleaved prM, and if its contribution is important, it would require to explicitly model its presence in their simulations.

Western blots showed that the VLPs used in these studies had (varying among preparations) 10-20% uncleaved prM. As we now state in the Materials and methods section, those particles were harvested from adherent cells after 2 days at 28°C. We found that harvesting after 5 days or switching to suspension culture led to a higher fraction of uncleaved prM, but we have not attempted a systematic study of the variables. The blots (two of which are shown in Author response image 1) were from preparations made for the experiments reported.

**Author response image 1. respfig1:** Western blots for VLPs of DV1 and DV3. A WNV VLP control is on the left. These are photographed from a lab notebook, hence the rulings in the background.

2) Another concern is regarding the location of the 3-110-22 binding site. The authors show that 3-110-22 and the fluorescently labeled analog compete for the same binding site, which is "presumed" to be the detergent binding pocket of the E protein. Based on the fusion simulation data and on the maximum number of fluorophore per VLP the compound is proposed to bind to the E protein dimer and inhibit the dimer to monomer transition as opposed to other steps downstream in the fusion reaction. Ultimately, a structure of the bound inhibitor would be of interest, but this is outside the scope of this Research Advance. In the absence of such a structure (or structure/function mutagenesis), a more in depth discussion of the likely binding site and likely mechanism of inhibition would be desirable.

A paper (from several of the co-authors of this manuscript) has now been accepted (at Cell Chem Biol) confirming, with mutation and with crosslinking studies (of a member of another series of compounds that competes with 3-110-22), that the binding site is indeed the β-OG pocket. We have cited this paper (in press) in the Introduction, and modified the final paragraph to discuss further its relevance to the data in our current manuscript.